

# Dissolution and precipitation of fractures in soluble rock

Georg Kaufmann[1], Franci Gabrovšek[2], and Douchko Romanov[1]

[1]Freie Universität Berlin, Institut für Geologische Wissenschaften, Malteserstr. 74-100, Haus D, 12249 Berlin, Germany
[2]Karst Research Institute ZRC SAZU, Postojna, Slovenia

*Correspondence to:* Georg Kaufmann (georg.kaufmann@fu-berlin.de)

**Abstract.** Soluble rocks such as limestone, anhydrite, and gypsum are characterised by their large secondary permeability, which results from the interaction of water circulating through the rock and dissolving the soluble fracture walls. This highly selective dissolution process enlarges the fractures to voids and eventually cavities, which then carry the majority of flow through an aquifer along preferential flow paths.

We employ a numerical model describing the evolution of secondary porosity in a soluble rock to study the evolution of isolated fractures in different rock types. Our main focus is three-fold: The identification of shallow versus deep flow paths and their evolution for different rock types; the effect of precipitation of the dissolved material in the fracture; and finally the complication of fracture enlargement in fractures composed of several different soluble materials.

Our results show that the evolution of fractures composed of limestone and gypsum is comparable, but the evolution time scale is drastically different. For anhydrite, owing to its difference in the kinetical rate law describing the removal of soluble rock, when compared to limestone and anhydrite, the evolution is even faster.

Precipitation of the dissolved rock due to changes in the hydrochemical conditions can clog fractures fairly fast, thus changing the pattern of preferential pathways in the soluble aquifer, especially with depth.

Finally, limestone fracture coated with gypsum, as frequently observed in caves, will result in a substantial increase in fracture enlargement with time, thus giving these fractures a hydraulic advantage over pure limestone fractures in their competition for capturing flow.

## 1 Introduction

Soluble rocks such as limestone, dolostone, gypsum, and anhydrite are characterised by the dissolution of material by water, often enriched with carbon dioxide. Removal occurs both on the rock surface and along fractures and faults in the subsurface in the case of a telogenetic origin of the rock. However, an aquifer in soluble fractured rocks evolves completely differently when compared to an aquifer in insoluble fractured rocks. While in the latter the permeability given by the continuous subsurface flow paths between interconnected fractures and bedding planes remains mainly constant, in soluble fractured aquifers permeability increases substantially with time. The reason for this strong increase in permeability is the enlargement of fractures and bedding partings by dissolution. The resulting flow paths in soluble fractured aquifers provide efficient drainage paths for water, and surface flow often completely disappears, as the enlarged fractures can swallow large amounts of water, even during larger



recharge events. The subsurface flow paths are, however, preferentially developed, and often voids enlarged to the meter size and more provide access to caves carrying entire underground rivers.

The evolution of voids and cavities has been described in the past by two competing hypotheses: *Water-table caves* evolving along a preferential flow path close to the water table (e.g. Swinnerton, 1932; Rhoades and Sinacori, 1941), or *bathyphreatic*

*caves* formed deeper below the water table along preferential flow paths (e.g. Davis, 1930; Bretz, 1942). Ford and Ewers (1978) have reconciled these two competing hypotheses with the definition of the *four-state model*: In this model, fracture spacing controls the type of cave evolution. For low fracture spacing, caves with deep phreatic loops can evolve, higher fracture spacing facilitates evolution of caves along the water table. Worthington (2001) came up with a different explanation of deep phreatic flow paths: Viscosity decreases with temperature, thus with depth, and therefore flow deeper in the aquifer can have a

slight hydraulic advantage over shallow flow (*hydraulic control*). Kaufmann et al. (2014), however, have shown by means of numerical modelling that the flow enhancement due to the decrease in viscosity with depth (*hydraulic control*) is counteracted by the reduction of solubility with depth (*chemical control*) and the reduction of initial fracture width with depth due to lithostatic pressure (*structural control*).

In this paper, we will go beyond the model developed by Kaufmann et al. (2014) for fractures in limestone by addressing

three key points: (i) Extend shallow and deep flow and evolution in fractures to more soluble rock types (limestone, gypsum, anhydrite), (ii) include precipitation into the evolution of the fractures, (iii) discuss the evolution of fractures composed of several soluble rock layers.

We have organised the paper as follows: In section 2, we discuss processes controlling the evolution of fractures. In section 3, we introduce physical and chemical principles for flow and evolution of a single fracture embedded in soluble rock. In section 4,

we discuss the modelling results, starting with the enlargement of single fractures in limestone, gypsum and anhydrite in shallow and deep conditions. We then proceed to describe changes due to possible precipitation of the soluble rock. We finally look at fractures composed of several soluble rock types. In section 5, we discuss and summarise our results.

## 2  Processes

We will discuss below mechanisms for fracture widening, fracture clogging, and the interaction of different dissolved species

in fractures.

### 2.1  Fracture widening

Fracture enlargement in soluble rocks is not a uniform process, but highly selective. Flow paths compete in the evolution and several factors can be responsible for preferential enlargement. We already introduced the four-state model of Ford and Ewers (1978), which is mainly controlled by structural differences, namely the fracture spacing in the field. Recently, the hypothesis of

*inception horizons* (e.g. Lowe, 1992, 2000; Filipponi and Jeannin, 2006; Filipponi et al., 2009) extended the idea of preferential enlargement of fractures and bedding planes. In this hypothesis, an inception horizon is identified as part of the soluble rock,





which has different physical, lithological or chemical properties than the surrounding rock. Often, inception horizons identified in the field are covered with gypsum.

## 2.2 Fracture dissolution and precipitation

The clogging of fractures can also play a key role in the development of flow through aquifers. Beside clogging through
sediments, large lithostatic pressure, or bacterial activity, soluble rocks can in places deliver solution supersaturated with a mineral species, which then starts to precipitate and reduces fracture width and thus flow. In the following, we review published work on fracture dissolution and precipitation with emphasis on the small spatial scale.

Bekri et al. (1997) investigated dissolution and deposition in a fracture by numerical means. They systematically searched through a parameter space guided by flow properties (Peclet number) and reaction properties (Damköhler number). A sequence
of deposition-dissolution cycles lasting several hundred hours each reveal periodic changes in flow velocity, until a clogging of the fractures stops the evolution.

Dijk and Berkowitz (1998) have modelled transport through fractures of different dimensions with different flow velocities assigned under laminar flow conditions. The transport equation used accounts for a source term representing dissolution or precipitation of dissolved matter. The rate law describing the speed on the reaction was assumed to be linear, and coefficients were
taken from the quartz system. Dijk and Berkowitz (1998) showed that precipitation occurs under a wide range of conditions. Dijk et al. (2002) examined the dissolution patterns in an artificial halite fracture in the laboratory. Using nuclear magnetic resonance imaging, they reported flow and density in the evolving fracture. Fracture enlargement was shown to be depending on the flow regime (Damköhler number) and a complicated function of flow and dissolution in the fracture.

Ross et al. (2001) have investigated a limestone fracture in the laboratory 50 cm long and with a cross-sectional area of
2.5 mm$^2$, and applied a hydraulic head difference resulting in a hydraulic gradient of 0.35. The resulting hydraulic conductivity measured was $K_f \simeq 340.6$ cm/min. After the injection of the circulating water with active groundwater microbes, the hydraulic conductivity of the fracture decreased by two orders of magnitude within 22 days due to the activity of the microbes.

Sigurindy and Berkovitz (2004) used field samples of dolomite to study flow, dissolution and precipitation of calcite in the laboratory. Depending on the residence time of the transferred water in the dolomitic fracture, either Aragonite precipitated
first (short residence times), then converted to calcite, or calcite precipitated directly (longer residence times).

Chaudhuri et al. (2008) modelled dissolution and precipitation in wide rectangular fractures, and showed that fractures starting from a non-uniform aperture width tend to develop preferential flow pathways in dissolution-dominated regimes, and become clogged by barriers perpendicular to flow in precipitation-dominated regimes.

Elkhoury et al. (2013) investigated the dissolution of calcite fractures under laboratory conditions with elevated temperature
and hydraulic pressure. The reported dissolution in the fractures becomes more uniform along the fracture length, when the flow rate through the fracture increases.

Noiriel et al. (2013) used calcite samples in the laboratory and injected $CO_2$-rich water, which starts dissolving the fracture walls. Monitoring flow and fracture geometry, they reported an increase in fracture roughness due to variations in dissolution





kinetics. The measured flow through the fracture deviates from idealized Navier-Stokes flow, but the deviation has only a small effect on the evolution of permeability on the macro-scale.

Jones and Dogwiler (2016) used two glass plates with rough surface, which where seeded with isolated regions of calcite. These fractures where then subject to flow with a $CaCl_2$-$NaHCO_3$-solution. The non-uniform and different mineral surface

fractures developed clogging around the parts originally covered with calcite, initiating preferential flow in the remainder of the fracture.

With the examples given above, we have highlighted the complicated development of fractures in soluble rock on the micro-scale. While most of the work discussed above is dealing with fracture dissolution and precipitation at the small scale, we will shift our focus to larger aquifer-size fractures in the following. Thus we simplify our numerical fracture to a circular

conduit with variable diameter, flow will be driven either under laminar or turbulent flow conditions, with a fracture roughness coefficient mimicking small-scale wall irregularities in the fracture, and transport of dissolved species as an advective process.

## 2.3 Anhydrite-gypsum conversion

Two further examples will motivate our work. As a first case the failed geothermal drill holes in Staufen (Breisgau, Germany) exemplifies the dissolution and precipitation in anhydrite-bearing layers. In 2007, seven boreholes for geothermal heat ex-

change were drilled in the city of Staufen to a depth of 140 m (e.g. Sass and Burbaum, 2010; Lubitz et al., 2013). One borehole connected two independent groundwater horizons and flow through an intermittent anhydrite lens caused widespread dissolution of the anhydrite and subsequent precipitation of gypsum. The larger volume of the precipitated gypsum caused surface uplift of up to 1 cm/month in an area 100×150 m, which caused widespread structural damage to the historical buildings.

Similar uplift rates related to geothermal drilling have been observed at Landau (Germany). Here, enlarged voids in around

450 m depth are likely to have been filled with injection fluids and caused uplift of $1 - 5$ cm/year (e.g. Heimlich et al., 2015). After the production well was shut down, the area subsided again.

## 3   Modelling flow and evolution

The evolution of fractures in soluble rock is controlled by both flow through the fracture and changes of the geometry of the fracture due to dissolution of and precipitation on the fracture walls. These two groups of properties, which we term *hydraulic*

properties and *chemical* properties, are coupled, and this coupling often results in a positive feedback mechanism responsible for the preferential enlargement of fractures and bedding partings in soluble rock.

In this section, we collect the relevant relations needed to describe and understand this feedback between hydraulic and chemical processes. In Fig. 1, a typical cross section 3 km long and 2 km deep for such a fracture is outlined: Two fracture pathes are shown (while with black outline): one flow path close to the surface along the water table, the other flow path resembling

bathyphreatic conditions and reaching into zones, where properties such as temperature, pressure and related parameters differ substantially from their surface values.





### 3.1 Hydraulic properties

We start by defining a fracture as a circular conduit of width $d$ [m] and length $l$ [m], and the deepest point of the fracture as $z_{max}$ [m]. Flow through this fracture can be described in general by the Navier-Stokes equations, which in our case can be simplified to classical Darcy flow under laminar or turbulent conditions (e.g. Beek and Muttzall, 1975):

$$Q = \begin{cases} Q_l & = & \frac{\pi \rho_w g}{128\eta} \frac{d^4}{l} \Delta h, \\ Q_t & = & \sqrt{\frac{\pi^2 g}{8f} \frac{d^5}{l} \Delta h}. \end{cases} \tag{1}$$

with $Q$ [m³/s] the flow rate through the fracture, $\Delta h$ [m] the drop in hydraulic head between both ends of the fracture, $\rho_w$ [kg/m³] the fluid density, $g$ [m/s²] gravitational acceleration, $\eta$ [Pa s] fluid viscosity, and $f$ [-] the friction factor. The linear relation between hydraulic head drop and flow, $Q_l$, describes laminar flow, while the quadratic relation, $Q_t$, represents turbulent flow conditions.

The friction factor depends on the dimensionless *Reynolds* number $Re$:

$$Re = \frac{\rho_w Q}{\eta} \frac{d}{A}, \tag{2}$$

with $A$ [m²] the cross-sectional area of the fracture. If the Reynolds number is below a certain threshold ($Re_c \sim 1000 - 2200$), flow in the fracture is laminar, above the threshold flow becomes turbulent. The friction factor can be defined as (e.g. Jeppson, 1976):

$$f = \begin{cases} f_l & = & \frac{64}{Re}, & Re \leq Re_c, \\ f_s & = & 0.3164 Re^{-0.25}, & Re \geq Re_c, \\ f_t^{-0.5} & = & 1.14 - 2\log\left(\frac{w}{d} + \frac{9.35}{Re} f_t^{-0.5}\right), & Re \geq Re_c, \\ f_r^{-0.5} & = & 1.14 - 2\log\left(\frac{w}{d}\right), & Re \geq Re_c, \end{cases} \tag{3}$$

with $w$ [m] the wall roughness, a parameter describing the roughness of the fracture walls. The different representations of $f$ describe laminar flow ($f_l$) and different stages of turbulent flow ($f_s$-smooth turbulence, $f_t$-transitional turbulence, $f_r$-rough turbulence). For the latter, the largest value controls turbulent flow behaviour. Note that both Reynolds number and friction factor have to be calculated with an iterative procedure.

The flowrate depends strongly on the conduit width (power-law of the order 3-4). The change in conduit width can be described (e.g. Kaufmann et al., 2014):

$$d(t_{i+1}) = d(t_i) + F \frac{m_{rock}}{\rho_{rock}} \Delta t, \tag{4}$$

with $t_i$ and $t_{i+1}$ [s] two consecutive time steps, $F$ [mol/m²/s] the calcium flux rate, $m_{rock}$ [kg/mol] the atomic mass of the dissolved rock, $\rho_{rock}$ [kg/m³] the density of the dissolved rock, and $\Delta t = t_{i+1} - t_i$ [s]. Note that $d(t=0) = d_0$ is the initial width of the fracture. The change in conduit width therefore depends on the calcium flux rate, with $F > 0$ describing enlargement of the fracture size by removing material from the fracture walls, and $F < 0$ reduction of the fracture width by


precipitating material on the fracture walls. The calcium flux rate itself is a function of the calcium concentration $c$ [mol/m$^3$] in the fracture. During dissolution, $c$ will increase along the fracture because of the removal of soluble rock from the fracture walls, until $c$ reaches the calcium equilibrium concentration $c_{eq}$ [mol/m$^3$]. The increase in calcium concentration along the fracture is calculated as

$$c(x_{i+1}) = c(x_i) + \frac{F(x)P(x)}{Q}\Delta x_i,\tag{5}$$

with $x_i$ and $x_{i+1}$[m] two neighboring points along the fracture, $\Delta x = x_{i+1} - x_i$, $F(x)$ [mol/m$^2$/s] the calcium flux rate, and $P(x)$ [m] the perimeter of the fracture. Note that $c(x_0) = c_{in}$ is the input calcium concentration $c_{in}$ [mol/m$^3$] entering the fracture.

## 3.2  Chemical properties

The dissolution or precipitation of material from a fracture in soluble rock depends on the amount of soluble material that can be dissolved from the rock wall into the solution flowing through the fracture. Depending on the type of soluble rock, the equilibrium concentration of the dissolved species is a function of several properties.

We will focus on gypsum (CaSO$_4$ · 2H$_2$O), anhydrite (CaSO$_4$), and limestone (CaCO$_3$) as soluble rock types. The chemical reactions for these different soluble rocks in an aqueous solution can be described through the following set of reactions (e.g. Duan and Li, 2008; Li and Duan, 2011):

$$
\begin{aligned}
\mathrm{H_2O} &\rightleftharpoons \mathrm{H^+ + OH^-} \\
\mathrm{H_2O + CO_2} &\rightleftharpoons \mathrm{H_2CO_3} \\
\mathrm{H_2CO_3} &\rightleftharpoons \mathrm{H^+ + HCO_3^-} \\
\mathrm{HCO_3^-} &\rightleftharpoons \mathrm{H^+ + CO_3^{2-}} \\
\mathrm{CaSO_4 + H_2O} &\rightleftharpoons \mathrm{Ca^{2+} + SO_4^{2-} + H_2O} \\
\mathrm{CaSO_4 \cdot 2H_2O} &\rightleftharpoons \mathrm{Ca^{2+} + SO_4^{2-} + 2H_2O} \\
\mathrm{CaCO_3 + H_2O} &\rightleftharpoons \mathrm{Ca^{2+} + CO_3^{2-} + H_2O} \\
\mathrm{CaCO_3 + H^+} &\rightleftharpoons \mathrm{Ca^{2+} + HCO_3^-} \\
\mathrm{CaCO_3 + H_2CO_3} &\rightleftharpoons \mathrm{Ca^{2+} + 2HCO_3^-}
\end{aligned}\tag{6}
$$

The first reaction describes the dissociation of water, the second the solution of carbon dioxide (CO$_2$) in water, the next two ones the dissociation of carbonic acid, and the remaining equations the solution of gypsum, anhydrite and limestone, respectively.

All of the above equations can be described as equilibrium reactions with their respective mass action coefficients, $K_i$. The calcium equilibrium concentration for the soluble rock types can be derived as approximate function to excellent accuracy (e.g.





Dreybrodt, 1988):

$$
\begin{aligned}
c_{eq} &= \sqrt[2]{\frac{K_A}{\gamma_{Ca^{2+}}\gamma_{SO_4^{2-}}}} && Anhydrite \\
c_{eq} &= \sqrt[2]{\frac{K_G}{\gamma_{Ca^{2+}}\gamma_{SO_4^{2-}}}} && Gypsum \\
c_{eq} &= \sqrt[3]{\frac{K_1 K_C K_H}{4K_2 \gamma_{Ca^{2+}}\gamma_{HCO_3^-}^2} p_{CO_2}} && Calcite
\end{aligned}
\tag{7}
$$

with $K_A(T,z)$ (Blount and Dickson, 1973) the equilibrium constant for the dissolution of anhydrite, $K_G(T,z)$ (Blount and Dickson, 1973) the equilibrium constant for the dissolution of gypsum, $K_H(T,z)$ (Weiss, 1974) the equilibrium constant for the

dissolution of atmospheric carbon dioxide into water (Henry's law constant), $K_0(T,z)$ (Wissbrun et al., 1954) the equilibrium constant for the reaction of water and carbon dioxide to carbonic acid, $K_1(T,z)$ and $K_2(T,z)$ (Millero, 1979; Mehrbach et al., 1973) the equilibrium constants for the dissociation of carbonic acid into bicarbonate, carbonate, and hydrogen, $K_C(T,z)$ (Mucci, 1983) the equilibrium constant for dissolved calcite, $\gamma_{Ca^{2+}}$, $\gamma_{SO_4^{2-}}$ and $\gamma_{HCO_3^-}$ the activity coefficients for calcium, sulfate and bicarbonate, $p_{CO_2}$ [atm] the carbon-dioxide partial pressure, and $T$ [°C] the temperature of the solution.

Our analytical expressions compare well with the calculation of the calcium equilibrium concentration derived for the full electro-neutrality condition, which we verified by comparing our $c_{eq}$ to results obtained from PHREEQC (Parkhurst and Appelo, 2013).

For limestone, (7) is valid for the open system, in which the solution is in contact with the atmosphere, and carbon dioxide will be replenished by further solution of $CO_2$ from the atmosphere. However, most fracture enlargement takes place under

closed-system conditions, and here the $CO_2$ is consumed and thus decreases with dissolution. In this case, the carbon-dioxide partial pressure is (Dreybrodt, 1988):

$$
p_{CO_2} = p_{CO_2}^{atm} - \frac{c_{eq}}{K_H\left(1 + \dfrac{1}{K_0}\right)}
\tag{8}
$$

with $p_{CO_2}^{atm}$ [atm] the initial carbon-dioxide partial pressure obtained in the atmosphere and the soil.

The calcium equilibrium concentrations for limestone, gypsum, and anhydrite are shown in Fig. 2 as functions of temper-

ature $T$ [°C] and carbon-dioxide pressure $p_{CO_2}$ [atm] for the case of limestone. Note the large difference in $c_{eq}$ between limestone $c_{eq} \sim 1-5$ mol/m$^3$, gypsum $c_{eq} \sim 15$ mol/m$^3$, and anhydrite $c_{eq} \sim 40$ mol/m$^3$. Also note that $c_{eq}$ for both limestone and anhydrite is a retrograde function of temperature, while for gypsum the temperature relation is prograde for temperatures below 30°C, retrograde above that temperature.

The calcium flux rate describes flux of dissolved species from and to rock surface per unit area and per time. It is controlled

by several potentially rate-limiting processes on the bedrock surface, e.g. the surface reaction at the mineral surface and the transport of the dissolved species in the solution. Flux rates have been measured experimentally for limestone (e.g. Plummer et al., 1978; Svensson and Dreybrodt, 1992; Eisenlohr et al., 1999), and for gypsum and anhydrite (e.g. James and Lupton, 1978; Gobran and Miyamoto, 1985; Lebedev and Lekhov, 1990; Jeschke et al., 2001; Jeschke, 2002), and for limestone have been predicted numerically (e.g. Buhmann and Dreybrodt, 1985a, b; Dreybrodt and Kaufmann, 2007; Kaufmann et al., 2010).



Flux rates $F$ [mol/m²/s] can be described as a piece-wise function of the calcium concentration $c$ with respect to the calcium equilibrium concentration $c_{eq}$ (e.g. Palmer, 1991):

$$F = k_i \left(1 - \frac{c}{c_{eq}}\right)^{n_i},$$ (9)

with $k_i$ [mol/m²/s] a rate coefficient, $c$ [mol/m³] the actual calcium concentration, $c_{eq}$ [mol/m³] the calcium equilibrium concentration, and $n_i$ [-] a power-law exponent. Mass transport by diffusion is implicitly accounted for by a reduction of $k_i$. For a summary of coefficients see table 1.

The calcium flux rates for limestone, gypsum, and anhydrite are shown in Fig. 3 as a function of calcium concentration $c$ for different temperatures $T$ and carbon-dioxide pressures $p_{CO_2}$ in the case of limestone. For both limestone and gypsum, the flux rates for dissolution are characterised by a linear decrease with increasing calcium concentration, until a material-dependent threshold ($c_s$) is reached. From then on, the flux rates decrease following a power law, thus are much slower. The reason for these non-linear flux rates at high calcium concentrations is the accumulation of impurities, which originate from insoluble material in the soluble host rock (e.g. clay). Note that for anhydrite, the flux rate is non-linear over the entire range for dissolution, which results in drastically different behaviour, as we will see later.

Once the calcium concentration $c$ passes the calcium equilibrium concentration $c_{eq}$, precipitation starts. Here, experimental data are scarce, and only the precipitation rates for limestone are based on laboratory work (Buhmann and Dreybrodt, 1985b; Dreybrodt and Buhmann, 1991). For precipitation rates of anhydrite and gypsum, we assumed a linear relation, but we also discuss potential non-linearities later.

### 3.3 Depth dependence

The material parameters density, gravity, and viscosity are functions of several variables, e.g. temperature and water pressure, which will be parameterised as depth $z$ [m]. While we can safely neglect the depth dependence of the gravitational acceleration over the depth range of even deep karst aquifers, temperature, hydrostatic and lithostatic pressure, density, and viscosity have to be parameterised according to the increase in depth:

$$
\begin{aligned}
T(z) &= T_0 + \left(\frac{dT}{dz}\right)_{geotherm} z, \\
p_w(z) &= \rho_w g z, \\
p_l(z) &= \rho_l g z, \\
\rho(z) &= \rho_0 \left[1 - \alpha T(z) + \kappa_T p_w(z)\right], \\
\log \frac{\eta(z)}{\eta_0} &= \frac{20 - T}{96 + T} \left[a_0 + a_1(20 - T)\right. \\
&+ \left. a_2(20 - T)^2 + a_3(20 - T)^3\right],
\end{aligned}
$$ (10)

with $T_0$ [°C] the annually-averaged surface temperature, $(dT/dz)_{geotherm}$ the geothermal gradient, with $\rho_w$ [kg/m³] the density of water, and $\rho_l$ [kg/m³] the density of rock, $\rho_0$ [kg/m³] the respective surface values, $g$ [m/s²] the gravitational attraction, $\alpha$ [1/K] the thermal expansivity of water (e.g. Jones and Schoonover, 2002), $\kappa_T$ [1/Pa] the compressibility of water (e.g. Jones and Schoonover, 2002), $\eta$ [Pa s] the viscosity of water, $\eta_0$ [Pa s] the reference value for the viscosity of water at surface pressure





and 25°C, $a_i$ polynomial coefficients for the viscosity (Kestin et al., 1978). Note that of course the water pressure depends on the induced flow, but for the depth-dependence of parameter values we use the hydrostatic pressure as simplification. A more detailed discussion of the depth-dependent properties can be found in Kaufmann et al. (2014).

### 3.4 Model implementation

The equations developed above have been implemented by the authors into a simple numerical code in Fortran90. The program performs the following steps:

1. Parameter values are read (e.g. $d_0$, $l$, $z_{max}$, $\Delta h$, $c_{in}$, $p_{CO_2}$, $T_0$, $\frac{dT}{dz}{}_{geotherm}$).

2. The fracture is discretised into $nx$-elements, and the fracture path is assigned, following $z = z_{max}[1 - (\frac{2x}{l} - 1)^2]$, assigning a parabolic flow path for depth.

3. A time stepping routine then calculates the evolution of the fracture. For each time step, a parcel of water enters the first fracture element with the pre-defined calcium input concentration $c_{in}$. Then for each sub-sequent fracture element, temperature, density, viscosity, and fracture width are calculated for the given depth according to (10).

4. The flow rate in the fracture element is calculated, based on (1) and depending on the flow regime, using the equivalent resistance formula for conduit elements in series.

5. With these parameters, the calcium equilibrium concentration $c_{eq}$ (7) and the calcium flux rate $F$ (9) are calculated in each fracture element.

6. Based on the flux rate and the time step $\Delta t$, the new width of the fracture element is calculated according to (4).

7. Finally, the calcium concentration is changed according to (5), and then passed to the next fracture element.

We now have assembled all relevant information on flow, evolution, and depth dependence of the hydraulic and chemical parameter values.

## 4   Results

In this section, we discuss results from numerical solutions of the flow and evolution equations for a single circular fracture.

### 4.1   Shallow versus deep settings

We first start to discuss the evolution of a single fracture in different soluble rocks and for two cases: The first case we term *water-table* fracture, as it is horizontal and close to the surface. The second case we term *deep-phreatic* fracture, as is extends to considerable depth $z_{max}$. The fracture considered has a length of $l = 2000$ m and an initial width of $d_0 = 0.5$ mm for the water-table fracture. Flow is driven from left to right, with a hydraulic head drop of $\Delta h = 10$ m, and the calcium concentration





of solution entering the fracture is $c_{in} = 0$ mol/m$^3$. As climatic variables, we assume a temperature of $T = 10°$C and a carbon-dioxide pressure of $p_{CO_2} = 0.05$ atm.

### 4.1.1 Limestone

The temporal evolution of a fracture in limestone is shown in Fig. 4. For the given temperature and carbon-dioxide pressure,
the calcium equilibrium concentration is around $c_{eq} \simeq 2.11$ mol/m$^3$.

–   We first focus on the water-table fracture:

The fracture evolves as a *classical* fracture in soluble rocks (e.g. Palmer, 1991; Dreybrodt, 1990; Dreybrodt et al., 2005; Kaufmann, 2002): The initial enlargement is focussed to the very first part of the fracture, where a typical funnel shape evolves (red solid lines). The calcium concentration increases rapidly over a distance of just a few meters to attain values
around 90% of the equilibrium value. Most of the fracture thus only slowly enlarges due to the high-order kinetic rate law active here. With time, the entrance funnel migrates further into the fracture, and first-order kinetics moves forward. Once the first-order kinetics reaches the exit of the fracture, the fracture starts enlarging almost uniformly over its entire length (blue dashed lines), and flow has become turbulent. The transition from high-order to first-order kinetics at the exit of the fracture characterises this two-fold evolution, and the time first-order kinetics reaches the exit is termed *breakthrough*
*time* in the literature (e.g. Dreybrodt, 1996). This breakthrough event occurs at around $T_B \sim 140,000$ years in the case of the water-table fracture in limestone.

–   We now discuss the evolution of the deep-phreatic fracture:

This fracture evolves in a very different manner than the water-table fracture. During the early evolution the entrance part is enlarged to a funnel shape as before, but with increasing depth the fracture evolution becomes inhibited. The reason for
the slow enlargement in greater depth is the decrease in calcium equilibrium concentration: At 50 m depth, the temperature is $T(50\ m) = 11.25°$C, hydraulic pressure is $p_w(50\ m) = 0.5$ MPa, and thus the calcium equilibrium concentration reduces to $c_{eq}(50\ m) \simeq 2.05$ mol/m$^3$, which is around 3% lower than the surface value. This lower calcium equilibrium concentration results in slower enlargement with depth, while the ascending part of the fracture (last half of the fracture) enlarges more, as here the calcium equilibrium concentration increases again and allows for faster dissolution (red solid
lines). With time, first-order kinetics will also be established in this case, and from then on a breakthrough event occurs, and after that the fracture grows with almost uniform enlargement rates. Note that we increased the initial fracture width to $d_i = 1.5$ mm for the deep-phreatic fracture to achieve a breakthrough time comparable to the water-table fracture (see Kaufmann et al., 2014, for further details).

Note that for a significant depth extent of the deep-phreatic fracture, the elevated temperatures in that depth may result
in a reduced calcium equilibrium concentration below the actual calcium concentration, thus precipitation would occur and the fracture will clog. We will discuss this clogging process later.





### 4.1.2 Gypsum

The temporal evolution of a fracture in gypsum is shown in Fig. 5. For the given temperature, the calcium equilibrium concentration is around $c_{eq} \simeq 15.35$ mol/m$^3$.

– We start again with the water-table fracture:

The evolution of the water-table fracture is essentially the same as in the case of limestone: A preferential enlargement to a funnel shape at the entrance during the early phase (red solid lines), with first-order kinetics only active in the first few tens of meters. The high-order kinetics active along the entire remaining part of the fracture ensures a bottleneck for flow, as the fracture width close to the exit remains low and thus flow out is limited. Once the first-order kinetics migrates through the fracture and reaches the exit, the breakthrough event occurs, and from then on the fracture enlarges at constant pace (blue dashed lines) under turbulent flow conditions. Note, however, that the breakthrough time for the water-table fracture in gypsum is with $T_B \sim 24,000$ years an order of magnitude smaller than in the case of limestone!

– Next we discuss the deep-phreatic fracture:

For the deep-phreatic fracture in gypsum we again have increased the initial fracture width to $d_{ini} = 1.6$ mm to obtain a similar breakthrough time ($T_B \sim 30,000$ years). The evolution of this deep-phreatic fracture is, however, very different from the evolution of a similar fracture in limestone: Before breakthrough, we can distinguish three different regions. (i) An area close to the entrance of the fracture (100-200 m into the fracture), were the first-order kinetics results in a funnel-shape enlargement of the entrance part. (ii) An area of relatively uniform growth (200-1500 m into the fracture), which has already enlarged to the centimeter-scale before breakthrough. (iii) The area around the exit of the fracture, which is still small, enlarging at a very slow pace and thus resulting in the bottleneck for flow responsible for keeping the calcium concentration in the high-order regime over large parts of the fracture. The reason for the different evolution before breakthrough is the prograde dependence of the calcium equilibrium concentration for gypsum for temperatures below 30°C: At 50 m depth, the calcium equilibrium concentration increases to $c_{eq}(30\ m) \simeq 15.61$ mol/m$^3$, which is around 2% larger than the surface value. Thus while the majority of the fracture experiences high-order kinetics before breakthrough, in the lower parts of the fracture the solution is slighly more aggressive due to the increase in $c_{eq}$. This is responsible for the widespread enlargement in the deeper parts of the deep-phreatic fracture.

After breakthrough is established, the entire fracture enlarges at almost constant pace (blue dashed lines) under turbulent flow conditions.

### 4.1.3 Anhydrite

The temporal evolution of a fracture in anhydrite is shown in Fig. 6. For the given temperature, the calcium equilibrium concentration is around $c_{eq} \simeq 45$ mol/m$^3$.

– Anhydrite water-table fracture:

At a first glance, the evolution of the fracture width is not too different from the two previous cases. Before breakthrough,

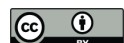



a funnel shape at the entrance part evolves, and the remainder of the fracture widens only very slowly (red solid lines). Once the breakthrough event occurs, the fracture grows at almost uniform pace (blue dashed lines) under turbulent flow conditions. However, two points are very different: (i) The breakthrough time is $T_B \sim 500$ years, which is very short. (ii) The calcium concentration profiles along the fracture differ from the previous two cases; while calcium concentration increases rapidly over the first few tens of meters, it remains below 90% of $c_{eq}$ for the remainder of the fracture for all times before breakthrough. This significant undersaturation is a result of the anhydrite calcium flux rate (see Fig. 3), which is *non-linear* for the entire dissolution branch. No first-order kinetics is present here, and thus the high-order kinetics keep the undersaturation with respect to calcium over the entire fracture length, and is therefore responsible for the fast evolution.

- Anhydrite deep-phreatic fracture:

  The above mentioned significant undersaturation with respect to calcium also changes the evolution for an anhydrite fracture reaching deeper into the subsurface. Here, no real difference from the water-table fracture evolution is observed, and breakthrough times for both flow paths are very similar. The drop in calcium equilibrium concentration due to the elevated temperature and water pressure at 30 m depth resulting in a calcium equilibrium concentration of $c_{eq}(30\ m) \sim$ 44.08 mol/m$^3$ and thus a reduction of around 2% relative to the surface value has no significant effect on the strong undersaturation in the fracture, the high-order kinetics of the calcium flux rate for anhydrite dominates the evolution by far.

We note, however, that at this point we neglected the precipitation of gypsum, which will occur in the anhydrite fracture due to the differences in calcium equilibrium concentration, which will of course change the fracture evolution. We come back to this point later.

## 4.2 Clogging of fractures

In this second part of the results section, we will look into the problem of clogging the fracture by precipitation of the corresponding mineral. We will focus on limestone and gypsum as soluble rocks, because anhydrite will be discussed in the next section.

We argue that the fracture is now subject to inflow with supersaturated solution with respect to calcium. We restructure the fracture by addressing a length of $l = 2000$ m and and initial width of $d_{ini} = 2.0$ mm, and applying the hydraulic head difference of $\Delta h = 10$ m.

### 4.2.1 Limestone

We first consider limestone as fracture material, and provide a solution to the fracture, which is slightly supersaturated with respect to calcium: $c_{in} = 2.13$ mol/m$^3$, $c_{eq} = 2.12$ mol/m$^3$. In Fig. 7, the evolution of this fracture is shown. The fracture width starts reducing along the entrance section of the fracture, as here the (small) super-saturation is largest. The deposition becomes smaller along the fracture, as the excess calcium in the solution is consumed. An inverse funnel shape appears, as more and



more calcite is deposited, and after $T_C \sim 856$ years the entrance part of the fracture is closed ($T_C$-clogging time) and flow through the fracture stops. Note that still the majority of the initial void volume is present, as the fracture has only been sealed off by a plug.

### 4.2.2 Gypsum

We now make up a fracture of gypsum, and feed a solution with $c_{in} = 15.37$ mol/m$^3$, thus slightly supersaturated when compared to the calcium equilibrium concentration of $c_{eq} = 15.35$ mol/m$^3$. This fracture closes essentially in the same manner as the limestone fracture, but just in a fraction of the time ($T_C \sim 9$ years).

### 4.3 Anhydrite/ gypsum conversion

In section 4.1, we have seen that an anhydrite fracture evolves very fast due to the non-linear kinetics of the anhydrite dis-
10 solution. However, once the calcium concentration in the anhydrite fracture reaches the calcium equilibrium concentration of gypsum ($\sim 15$ mol/m$^3$), gypsum starts to precipitate:

$$
\begin{aligned}
\mathrm{CaSO_4 + 2H_2O} &\rightleftharpoons \mathrm{Ca^{2+} + SO^{2-} + 2H_2O} \\
&\rightleftharpoons \mathrm{CaSO_4 \cdot 2H_2O}
\end{aligned}
\tag{11}
$$

We have seen in section 4.2 that a gypsum fracture can become clogged by precipitation of gypsum in only a couple of years, even for small super-saturation with respect to gypsum.

In Fig. 8, the change in fracture width (wall retreat) is shown for a system at $T = 10°$C and both anhydrite and gypsum. The calcium concentration range chosen focuses around the calcium equilibrium concentration for gypsum. While the enlargement of a fracture in anhydrite with a chemical composition around $c \sim 15$ mol/m$^3$ is with $\Delta r \simeq 6 - 8$ mm/yr rather constant, the situation for precipitation of gypsum dramatically changes the evolution. Once the calcium equilibrium concentration for gypsum ($c_{eq}^{gypsum} = 15.3$ mol/m$^3$) is passed, gypsum will start to precipitate. If the calcium concentration in the fracture exceeds
$c_{eq}^{gypsum}$ just slightly, the deposition rate of gypsum quickly reaches values larger than $\Delta r > -10$ mm/yr, thus outpacing the removal of anhydrite by far. The fracture will clog very soon along the entrance part, and flow is inhibited. The reason for this quick clogging of course is the difference in atomic mass and density for anhydrite and gypsum, which translates into these different retreat rates according to (4).

As we have stated in section 3, the precipitation flux rates for both anhydrite and gypsum are not well known. We therefore
plotted the deposition rate for gypsum also for non-linear flux rate laws ($n_1 = 1.5$, dashed red line; $n_1 = 2$, dotted red line). We observe a fairly substantial impact of the non-linearity to gypsum precipitation, which, however, will still outpace the dissolution of anhydrite.

We speculate that this behaviour will have an impact on problems as during the drilling of the hydrothermal drill hole in Staufen, which connected two aquifers and caused flow of under-saturated solution through an anhydrite lens. As the anhydrite
gets dissolved, the calcium concentration reaches the threshold, when gypsum starts to precipitate. If the gypsum precipitate is able to get firmly attached to the fracture wall, clogging of the anhydrite fracture can occur. However, in the case of very high




flow conditions, the gypsum precipitate might be flushed out, before it is attached to the fracture wall, thus the fracture remains open.

## 4.4 Multi-material fractures

In this last section, we pick up the discussion on inception horizons from the introduction. There are numerous observations of faults and bedding planes in limestone more favorable to karstification than others (e.g. Filipponi et al., 2009; Plan et al., 2009, and references therein). Often, these inception horizons have been covered with pyrite ($FeS_2$), which then oxidised according to (Ritsema and Groenenberg, 1993):

$$
\begin{aligned}
4FeS_2 + 15O_2 + 14H_2O &\rightleftharpoons 4Fe(OH)_3 + 16H^+ + 8SO_4^{2-} \\
CaCO_3 + 2H^+ + SO_4^{2-} + H_2O &\rightleftharpoons CaSO_4 \cdot 2H_2O + CO_2
\end{aligned}
\tag{12}
$$

The first reaction describes the oxidation of pyrite. The latter reaction occurs at the boundary of the fracture wall, where the sulfate reacts with calcite to form gypsum. This gypsum, which covers the fracture, can then readily dissolve and provide a rapid initial enlargement of the limestone fracture. Note that the carbon dioxide released can then be dissolved in water and thus increase the calcium equilibrium concentration, water becomes again undersaturated with respect to calcite and can dissolve additional limestone.

We show such an example in Fig. 9. Here, a limestone fracture with a length of $l = 2000$ m and an initial width of $d_{ini} = 0.5$ mm is covered with a gypsum layer of 5 mm thickness. Flow is driven from left to right by the hydraulic head difference of $\Delta h = 10$ m, the incoming solution is aggressive ($c_{in} = 0$ mol/m$^3$).

The fracture experiences a two-stage evolution. First, the gypsum starts dissolving up to its saturation ($c_{eq}^{gypsum} \simeq 15$ mol/m$^3$). The entrance enlarges as funnel shape, the remainder of the fracture only slowly enlarges. After 10,000 years, the gypsum along the entrance section of the fracture is gone, the fracture here starts evolving by dissolving limestone with its lower saturation ($c_{eq}^{calcite} \simeq 2$ mol/m$^3$). As the solution saturated with respect to limestone is still aggressive with respect to the gypsum, the fast gypsum dissolution is pushed further into the fracture.

This scenario is similar to the *limited dissolution* proposed by Romanov et al. (2010) and Gabrovšek and Stepišnik (2011). If there is limitation of soluble material in the direction perpendicular to flow, enlargement becomes more uniform along the entire fracture. In our case, after 10,000 years the gypsum vanishes along the entire fracture, flow rate increases and dissolution of the exposed limestone then accelerates.

When we compare breakthrough times of this gypsum/limestone fracture ($T_B \sim 21,000$ years) to that of a similar fracture only composed of limestone ($T_B \sim 140,000$ years, see Fig. 4), we find an acceleration of evolution by an order of magnitude. This faster evolution can explain the importance of an inception horizon, which is chemically distinct from the other fractures by the gypsum cover. The inception horizon evolves way faster than the surrounding fractures and thus captures flow and provides a preferential pathway.

The limited dissolution due to the thin thickness of the gypsum cover plays an important role. We have reduced for the example fracture discussed above the thickness of the gypsum cover from 5 to 1 mm, and breakthrough time becomes even




shorter ($T_B \sim 13,000$ years). Thus two mechanisms can be identified as important for preferential karstification along inception horizons: The faster dissolution of the gypsum precipitate, and the more uniform enlargement due to the limited dissolution effect.

We can of course also speculate about the opposite effect: A limestone fracture coated with gypsum can clog, if the gypsum
layer has a non-uniform thickness. If a thinner gypsum coating within the fracture is removed first and the limestone exposed in that part, the solution arriving is significantly supersaturated with respect to limestone, and limestone will precipitate and can eventually clog flow of the entire fracture.

## 5  Conclusions

We have developed a formal framework for describing the evolution of a single fracture in a soluble rock, which can be enlarged
by dissolution of the rock, and which can clog as a result of precipitation of a mineral in solution. The single fracture can consist of limestone, gypsum, or anhydrite, or a combination of these soluble rock types.

A first result, already established in the literature, is the strong dependence of the evolution time on the type of soluble rock considered: While the enlargement of fractures in limestone under natural hydraulic conditions can be between 1000 and 10000 years, fractures in anhydrite and gypsum evolve much faster, on timescales of 10-100 years.

Deeper flow paths as considered in the bathyphreatic evolution of caves depend on the change of environmental parameters with depth: Temperature, water and rock pressure generally increase with depth, changing hydraulic properties (e.g. water viscosity), but also the chemical properties (e.g. calcium equilibrium concentration). Here, the retrograde dependence of calcium equilibrium concentration on temperature for limestone and anhydrite are responsible for a reduction of enlargement with depth, thus bathyphreatic flow paths evolve slower, when compared to water-table flow paths. For gypsum, however, the
prograde dependence of calcium equilibrium concentration on temperature (at least for temperatures below 30°C) does not inhibit evolution with depth.

Once solution becomes supersaturated with respect to calcium, either through changes in temperature and/or water pressure or evaporation, the fracture can clog due to precipitation. As the precipitation of soluble rock is largest along the inflow part of the fracture, clogging creates a plug inhibiting flow, but keeps large parts of the fracture further downstream still open.

If a fracture consists of more than one material, the evolution becomes more complicated. In our example of a limestone fracture coated with gypsum (e.g. from the conversion of pyrite into gypsum) evolves more quickly, when compared to a pure limestone fracture. Here, the gypsum coating is quickly removed along the entire fracture, thus the remaining limestone part has a larger diameter, which for the ongoing evolution exerts a strong control on the time of evolution.

*Author contributions.*



*Acknowledgements.* GK acknowledges funding from the DFG under research grant KA1723/6-2. This project has been carried out at the Karst Research Institute of Slovenia (ZRC SAZU) during a sabbatical stay of GK. Figures were prepared using GMT software (Wessel and Smith, 1998). Results of the numerical simulations can be obtained from GK upon request.



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



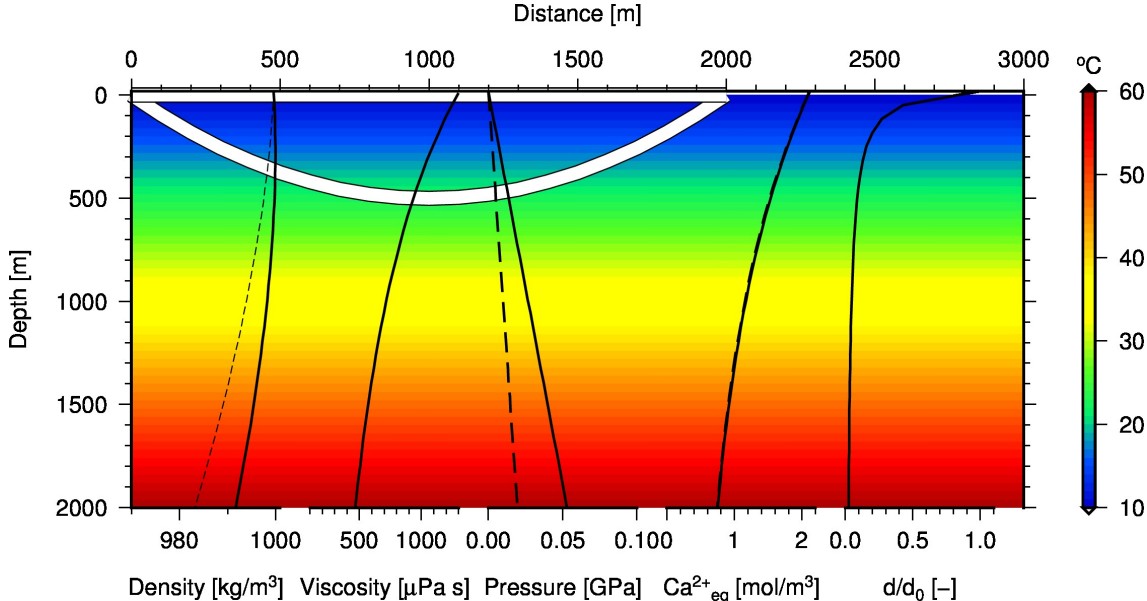

**Figure 1.** Model setup for the karst conduit evolution. Temperature is shown as color contours, ranging from $10°C$ at the surface to around $60°C$ in 2 km depth. The corresponding properties of water are shown as black lines: density $\rho$ (dashed line temperature effect only; solid line temperature and pressure effect), viscosity $\eta$, pressure $p$ (dashed line water pressure, solid line lithostatic pressure), calcium equilibrium concentration $c_{eq}$ (dashed line temperature effect only, solid line temperature and pressure effect; not distinguishable), fracture-depth relation $d/d_i$. The thick white lines with black outline indicate two conduit paths, one along the water table, one deep bathyphreatic cave.




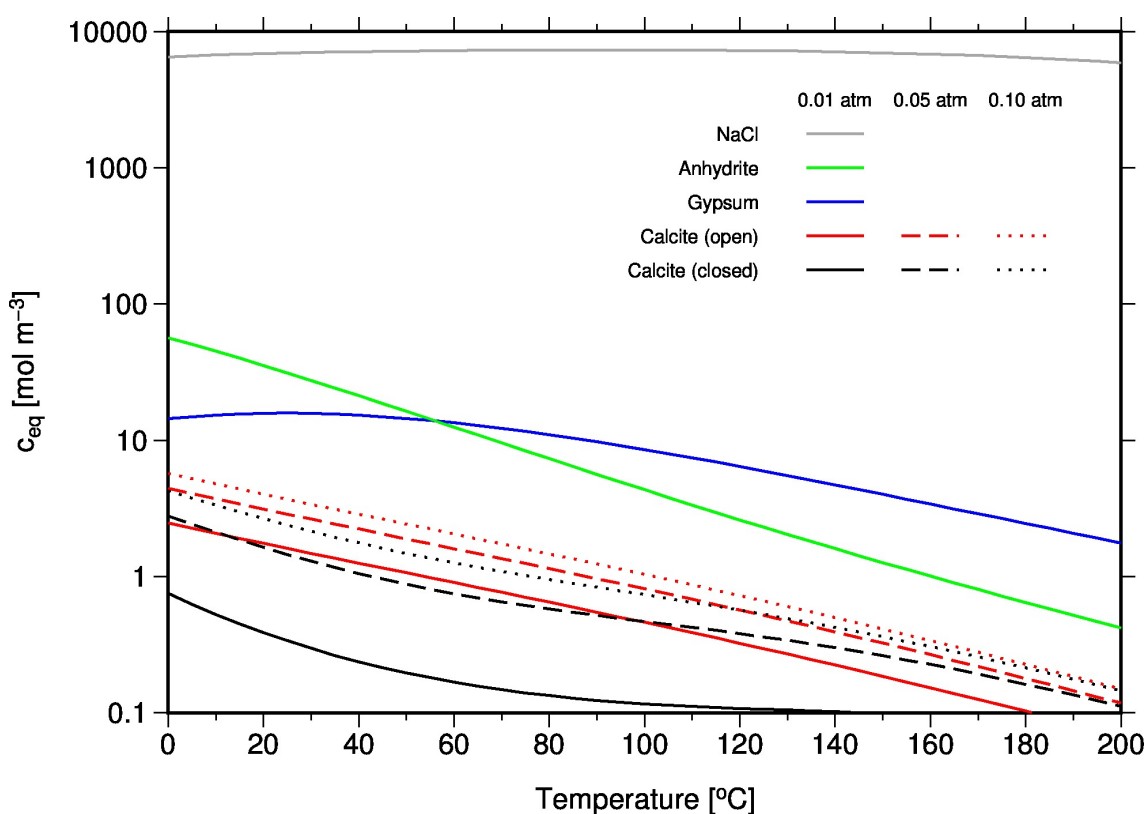

**Figure 2.** Calcium equilibrium concentration as a function of temperature. Shown are curves for limestone ($CaCO_3$), gypsum ($CaSO4{\cdot}H_2O$), anhydrite ($CaSO_4$), and halite ($NaCl$). For limestone, the calcium equilibrium concentration is shown for three values of partial carbon-dioxide pressures and for open- and closed-system conditions.





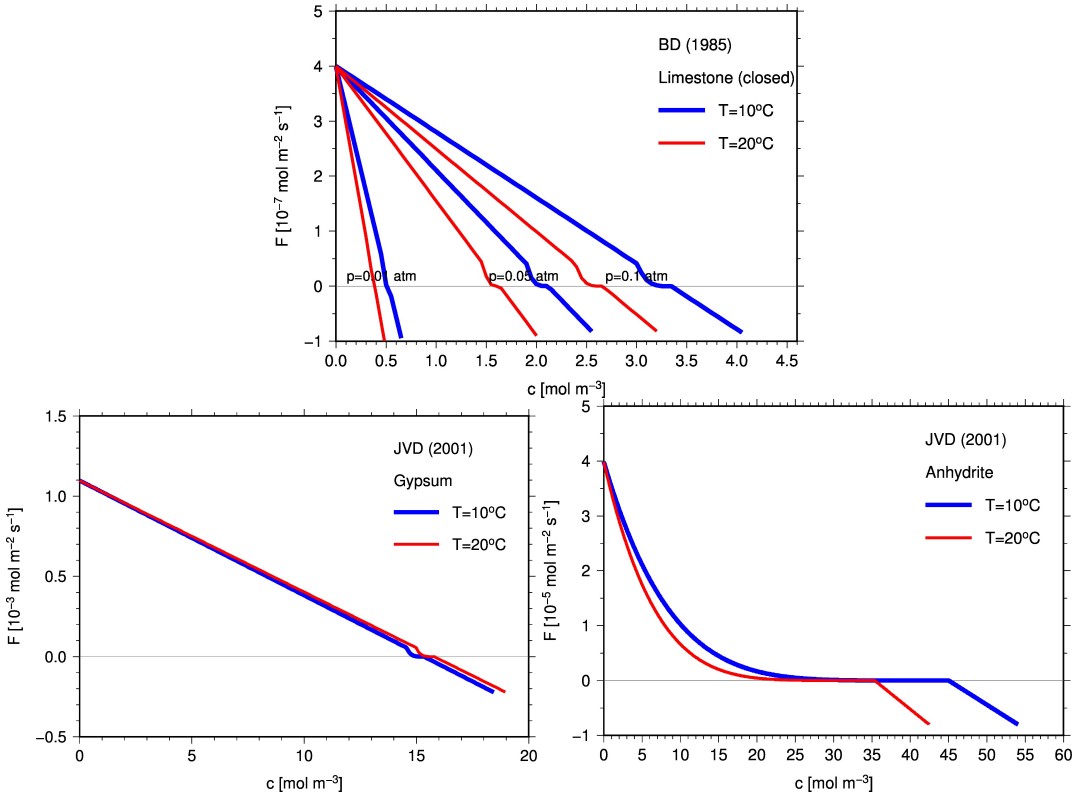

**Figure 3.** Calcium flux rates as a function of calcium concentration for different soluble rocks. Curves are shown for two different temperatures and in the case of limestone also for three different partial carbon-dioxide pressures. Top: limestone, bottom left: gypsum, bottom right: anhydrite.



**Figure 4.** Evolution of single limestone fracture with time. Shown are fracture width and calcium concentration for several time steps (see notation on lines in years). Red lines indicate period before breakthrough, blue lines after breakthrough, and the flow condition is either laminar (solid lines) or turbulent (dashed lines). Top: Limestone water-table fracture. Bottom: Limestone deep-phreatic fracture.





**Figure 5.** As Fig. 4, but for gypsum fracture.





**Figure 6.** As Fig. 4, but for anhydrite fracture.





**Figure 7.** Clogging of single fracture with time by precipitation. Note that only the first 200 m of the fractures is shown. Evolution of width and calcium concentration for limestone fracture (top) and for gypsum fracture (bottom).




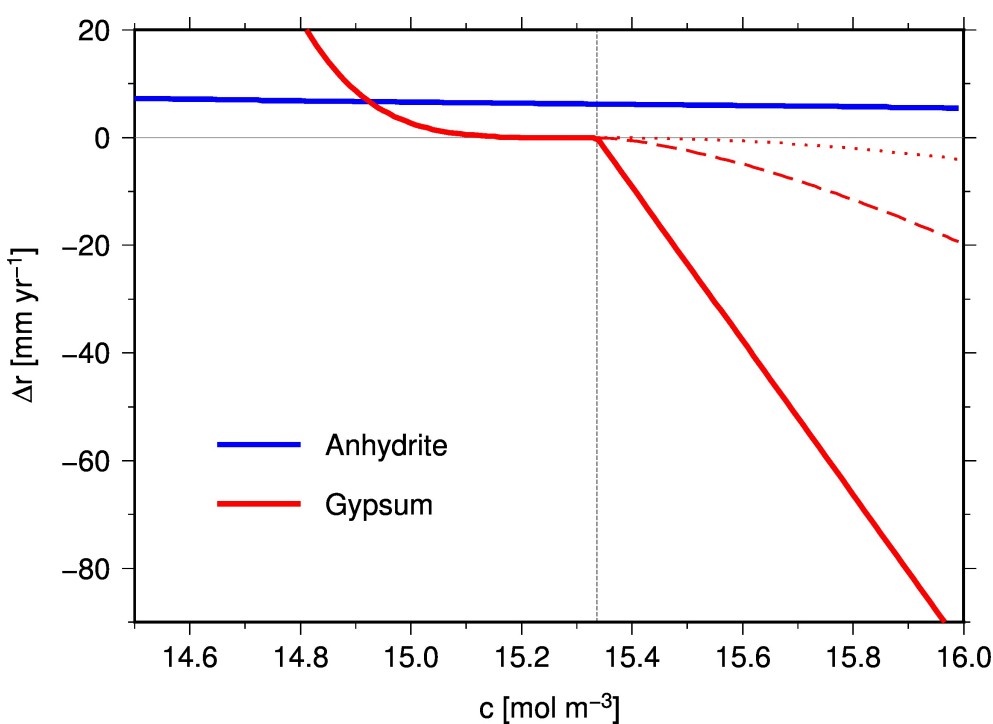

**Figure 8.** Wall retreat $\Delta r$ as a function of calcium concentration $c$ for anhydrite (blue solid line) and gypsum (red lines). For gypsum, different rate-equation exponents for precipitation are shown ($n_1 = 1.0$, solid, $n_1 = 1.5$, dashed, $n_1 = 2.0$, dotted). The dashed grey line marks the calcium equilibrium concentration for gypsum at $T = 10°$C.





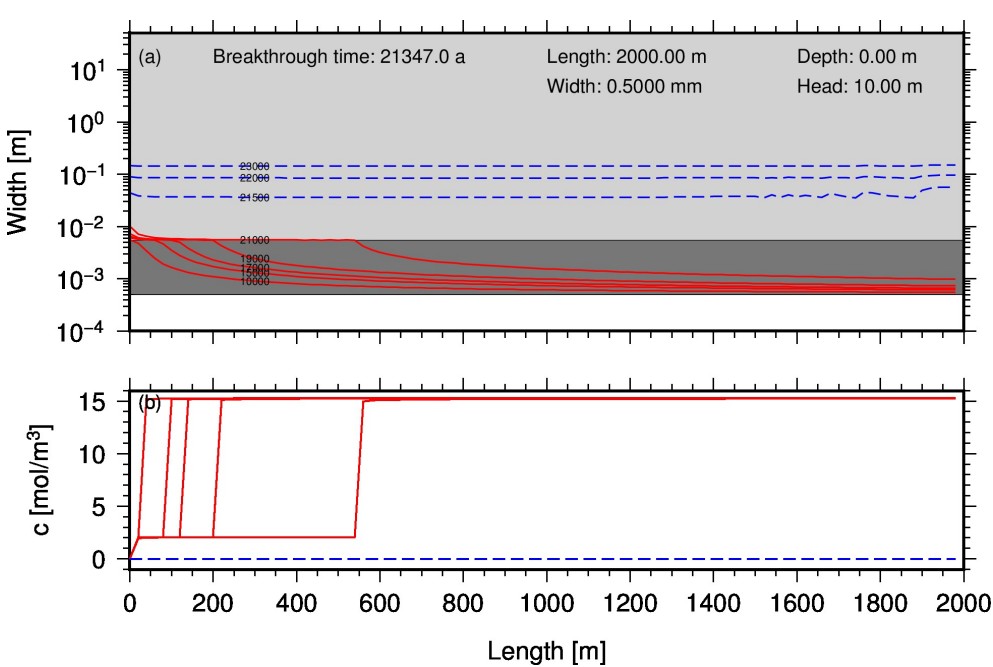

**Figure 9.** Evolution of limestone fracture with gypsum coating.





**Table 1.** Parameter values for soluble rock chemistry.

| | | | | |
|---|---|---|---|---|
| Limestone | atomic mass | $m_{Rock}$ | [kg/mol] | 0.100 |
| | density | $\rho_{Rock}$ | [kg/m$^3$] | 2600 |
| | rate constant[1] | $k_1$ | [mol/m$^2$/s] | $4 \times 10^{-7}$ |
| | rate exponent[1] | $n_1$ | [-] | 1.0 |
| | rate exponent[1] | $n_2$ | [-] | 4.0 |
| | switch $c_{eq}$[1] | $c_s$ | [-] | 0.90 |
| Anhydrite | atomic mass | $m_{Rock}$ | [kg/mol] | 0.136 |
| | density | $\rho_{Rock}$ | [kg/m$^3$] | 2900 |
| | rate constant[2] | $k_1$ | [mol/m$^2$/s] | $4.0 \times 10^{-5}$ |
| | rate exponent[2] | $n_1$ | [-] | 5.4 |
| Gypsum | atomic mass | $m_{Rock}$ | [kg/mol] | 0.172 |
| | density | $\rho_{Rock}$ | [kg/m$^3$] | 2200 |
| | rate constant[3] | $k_1$ | [mol/m$^2$/s] | $1.1 \times 10^{-3}$ |
| | rate exponent[3] | $n_1$ | [-] | 1.0 |
| | rate exponent[3] | $n_2$ | [-] | 4.5 |
| | switch $c_{eq}$[3] | $c_s$ | [-] | 0.95 |

[1] Buhmann et al. (1985)

[2] Jeschke (2002)

[3] Jeschke et al. (2001)

**Table 2.** Parameter values for depth dependences of material properties.

| | | |
|---|---|---|
| Geothermal gradient | $(dT/dz)_{geotherm}$ | 25 $^\circ$C/km |
| Thermal expansivity | $\alpha$ | |
| Compressibility | $\kappa_T$ | |
| Viscosity | $\eta_0$ | $1.002 \times 10^{-3}$ Pa s |
| | $a_0$ | 1.2378 |
| | $a_1$ | $1.303 \times 10^{-3}$ |
| | $a_2$ | $3.060 \times 10^{-6}$ |
| | $a_3$ | $2.550 \times 10^{-8}$ |





**Table 3.** Parameter values for standard model.

| | | |
|---|---|---|
| Fracture length | $l$ | 2000 m |
| Fracture radius | $d$ | 0.5 mm |
| Max. fracture depth | $z_{max}$ | 0 m |
| Head drop | $\Delta h$ | 10 m |
| Surface temperature | $T_0$ | 10°C |
| $CO_2$-pressure | $p_{CO_2}$ | 0.05 atm |
| Calcium input concentration | $c_{in}$ | 0 mol/m$^3$ |