# Peer review of "Dissolution and precipitation of fractures in soluble rock"

_Hydrology and Earth System Sciences, 2016_

## Referee Comment (RC1) · Anonymous Referee #1 · 20 Sep 2016

This manuscript presents results from reactive transport simulations in single fractures idealized as long one-dimensional tubes. Two different fracture configurations were considered: shallow fracture that remained near the water table over its entire length, and a deep fracture that followed a parabolic path from near the surface to a depth of $z_{max}$ and back to the surface. The study extends the results of Kaufmann et al. (2014) by simulating several mineralogies. In addition to limestone, the current study includes gypsum, anhydrite and gypsum-lined fractures in limestone. I have some general comments about the manuscript, an a number of detailed comments.

**General comments**

The motivation for the study outlined in the Introduction relies mostly on previous work of the authors and ignores the large amount of work over the last 15 years aimed

at better understanding permeability changes in fractured rock caused by fluid-mineral reactions. A sub-sample of other studies are cited in the Processes section, but each is briefly addressed independently with no effort to synthesize the results and findings of these previous studies to offer a compelling motivation for the current study. The statement that the previous studies focused on small-scale processes and the current study focuses on large-scale behavior is a bit of a generalization and not exactly true (e.g., Hanna and Rajaram, 1998; Chaudhuri et al., 2008; Szymczak and Ladd, 2011 each consider scaling issues associated with related reaction processes). This manuscript would be more compelling if it started with a more nuanced discussion of the motivation for the study in light of the significant number of related studies in the recent literature.

Some of the details of the model are not well justified or documented, which undermines the impact of the simulation results. For example, the decision to represent fractures as a single, one-dimensional tube is not supported by recent experimental, numerical, and theoretical results (see for example references cited in the previous paragraph), which show the importance of the two- or three-dimensional flow field within fractures on the development of preferential flow paths by dissolution. Here, you essentially assume that, at time zero, a preferential flow path exists across the entire domain and it then grows by dissolution. It may be reasonable to ignore the development stage of these preferential flow paths, but the onus is on you to explain why in the context of other papers that focus on this interesting process. See other modeling issues in my detailed comments below.

**Detailed comments**

p.4 lines 19-21: This example doesn't seem relevant. My understanding is that the uplift resulted from over-pressurized fluids, not from mineral precipitation.

p. 5 eq.1: Doesn't seem necessary to define $Q_l$ and $Q_t$ because your expression for $Q_t$ is also valid for laminar flow when the friction factor is defined as 64/Re (e.g., eq. 3).

p. 8 lines 5-6: This needs more discussion. It is true that mass transport across the diameter will reduce value of the effective reaction-rate coefficient, the amount of reduction depends on the fluid velocity, diffusion coefficient, and reaction rate (e.g., Szymczak and Ladd, 2011). Furthermore, when the flow transitions to turbulence, mass transport is no longer limited by diffusion, but turbulent mixing.

p. 8 lines 10-13: This deserves a reference.

p. 9 lines 1-2: Why simplify to assume a hydrostatic pressure distribution when the model implicitly calculates the pressure loss along the flow conduit? Calculating the actual pressures seems trivially easy.

p. 9 lines 13-14: How is the flow rate in each fracture element calculated? Presumably this involves solving a system of linear / nonlinear equations depending on the flow rate. More information on the details of these calculations would be helpful.

p. 9 lines 15-16: This must also include eq. 5 for the mass flux between each cell?

p.9 lines 10-18: The scheme described here is first-order in both space and time and, thus, likely quite sensitive to $\Delta t$ and $\Delta x$. Did you conduct a sensitivity analysis to ensure that $\Delta t$ and $\Delta x$ were adequately small to achieve convergence?

p. 10 line 7: What is a "classical fracture"?

p.10 line 27: Why increase the tube diameter for the deeper fracture? As noted by Kaufmann et al. (2014) and many others, fracture permeability is expected to decrease with depth. It would seem more physically relevant to include a tube with the same diameter at the surface and decreasing with depth than a larger fixed-diameter tube.

p. 12 line 25: Can you give an example when one might expect to have such a super-saturated influent solution?

---

## Referee Comment (RC2) · Anonymous Referee #2 · 20 Sep 2016

This manuscript entitled "Dissolution and precipitation of fractures in soluble rock" reports a numerical study of fracture evolution caused by mineral dissolution and precipitation. A 1D reactive transport model was developed to investigate the temporal evolution of fractures at different depths and in different soluble rocks. The study showed that fracture opening caused by mineral dissolution is highly dependent on the kinetical rate laws, and fracture clogging can be caused by precipitation of the dissolved mineral or secondary mineral in the downstream of the fracture. The study also showed that the evolution of fractures can be expedited by the presence of a coating or filling layer of different minerals.

This manuscript addresses scientific questions that are within the scope of HESS. The model framework and main findings are of interest to a broad audience. However, there are some issues to be addressed.

Major comments:

1. The introduction and literature review appear to be disconnected from the main focuses of this study. For example, the paragraph that starts with line 3 on page 2 elaborates on several modeling work, but it is unclear how this study is related to or different from those studies, except for the Kaufmann et al (2014) study. Moreover, while the literatures summarized in section 2.2 showed how fracture evolution is affected by different flow regimes and other factors, most of these factors are not addresses or are simplified in this study. For example, a drop of hydraulic head of 10 m was used in the manuscript, but it was not compared with realistic cases, and it is unclear why this specific flow regime is chosen. Furthermore, these literatures focus on small scale processes, while this study investigates field scale phenomena. Can the authors comment on how these literatures are relevant (or irrelevant) to this study and the scaling issue?

2. Some aspects of the model framework and parameterization need to be clarified.
   a. Page 4 line 10: 'with a fracture roughness coefficient mimicking small-scale wall irregularities in the fracture', is this roughness coefficient used in the calculation of reaction rate or flow or both?
   b. Page 5 line 13: what is the threshold Re used in this study?
   c. Page 5: $f_l$ the friction factor for laminal flow was presented in eqn(3) but not used in eqn (1), was it used at all?
   d. Page 5: how is the wall roughness (w in eqn(3)) defined and determined in this study and what is the impact of this parameter?
   e. Page 6: eqn (5) is very different from the advection-diffusion-reaction equation, even if the diffusion term is excluded. Can the authors comment on this and clarify the underlying assumptions? For example
      i. Is steady-state assumed, although it appears not to be the case given the following results?
      ii. Is CFL criterion assumed to be one?
   f. Page 8/29 (table 1):
      i. Only one kinetic coefficient is reported for the calcite reaction, but three reaction pathways were listed in (6), can the authors comment on this discrepancy?
      ii. For the gypsum reaction, the kinetic coefficients for the linear and non-linear rate laws are about one order of magnitude different according the reference

cited, but the authors used the same kinetic coefficient, why and how the results may be affected?

iii. The texts pointed out that different parameters are used in precipitation from dissolution, it should be clarified in table one.

g. Page 9 line 17: step 6 what is the time step?

h. Given the strong dependence of the evolution profile on kinetic rate laws, some sensitivity analysis or discussion of the uncertainties of the kinetic laws and coefficients should be provided.

Minor comments:

1. Page 2: how is section 2.1 fracture widening different from fracture dissolution discussed in section 2.2?

2. There are a couple of typos. For example, Page 4 line 3 should be 'Jones and Detwiler (2016)', and 'where' should be 'were'

---

## Author Comment (AC1) · 30 Sep 2016

Reply to interactive comment by *Anonymous Referee #1*

We would like to thank the Anonymous Referee #1 for his valuable comments and take the opportunity to discuss the points made.

**General comments**

The motivation for the study outlined in the Introduction relies mostly on previous work of the authors and ignores the large amount of work over the last 15 years aimed at better understanding permeability changes in fractured rock caused by fluid-mineral reactions. A sub-sample of other studies are cited in the Processes section, but each is briefly addressed independently with no effort to synthesize the results and findings of these previous studies to offer a compelling motivation for the current study. The

statement that the previous studies focused on small-scale processes and the current study focuses on large-scale behavior is a bit of a generalization and not exactly true (e.g., Hanna and Rajaram, 1998; Chaudhuri et al., 2008; Szymczak and Ladd, 2011 each consider scaling issues associated with related reaction processes).

We will move the review of selected work on fracture evolution to the introduction, group them into laboratory studies and numerical studies, and summarise their main outcomes as a motivation for our work. We note that the literature review is aimed to put our specific work into a broader context.

This manuscript would be more compelling if it started with a more nuanced discussion of the motivation for the study in light of the significant number of related studies in the recent literature. Some of the details of the model are not well justified or documented, which undermines the impact of the simulation results. For example, the decision to represent fractures as a single, one-dimensional tube is not supported by recent experimental, numerical, and theoretical results (see for example references cited in the previous paragraph), which show the importance of the two- or three-dimensional flow field within fractures on the development of preferential flow paths by dissolution. Here, you essentially assume that, at time zero, a preferential flow path exists across the entire domain and it then grows by dissolution. It may be reasonable to ignore the development stage of these preferential flow paths, but the onus is on you to explain why in the context of other papers that focus on this interesting process. See other modeling issues in my detailed comments below.

Our decision to use a single conduit to show dissolution and precipitation in a fracture for different soluble rock types is based on the assumption that the three aspects we choose to discuss with our work are best discussed in a simple setup. While we admit that reality is much more complicated, we would like to focus on the effect of chemical kinetics and rock type on the fracture evolution, keeping 2- and 3D-effects from fracture network out of the scope.

**Detailed comments**

p.4 lines 19-21: This example doesn't seem relevant. My understanding is that the uplift resulted from over-pressurized fluids, not from mineral precipitation.
The referee is correct, we will drop this example.

p. 5 eq.1: Doesn't seem necessary to define Ql and Qt because your expression for Qt is also valid for laminar flow when the friction factor is defined as 64/Re (e.g., eq. 3).
We redefine flow to a single non-linear equation.

p. 8 lines 5-6: This needs more discussion. It is true that mass transport across the diameter will reduce value of the effective reaction-rate coefficient, the amount of reduction depends on the fluid velocity, diffusion coefficient, and reaction rate (e.g., Szymczak and Ladd, 2011). Furthermore, when the flow transitions to turbulence, mass transport is no longer limited by diffusion, but turbulent mixing.
The modeling of calcium flux is a well established field, which has been well documented during the past. We add references to guide the reader to the relevant discussion.

p. 8 lines 10-13: This deserves a reference.
References will be added.

p. 9 lines 1-2: Why simplify to assume a hydrostatic pressure distribution when the model implicitly calculates the pressure loss along the flow conduit? Calculating the actual pressures seems trivially easy.
Due to the extent of the deeper models the hydrostatic pressure dominates, and hydraulic pressure is smaller. We will discuss our choice in the text now.

p. 9 lines 13-14: How is the flow rate in each fracture element calculated? Presumably this involves solving a system of linear / nonlinear equations depending on the flow rate. More information on the details of these calculations would be helpful.
We will add references to clarify our modeling approach.

p. 9 lines 15-16: This must also include eq. 5 for the mass flux between each cell?

Link to equation will be added.

p.9 lines 10-18: The scheme described here is first-order in both space and time and, thus, likely quite sensitive to $\Delta T$ and $\Delta x$. Did you conduct a sensitivity analysis to ensure that $\Delta T$ and $\Delta x$ were adequately small to achieve convergence?
We discretise the fracture with concentration increments, thus use the change in concentration to determine the length of each sub-element in the fracture. Time steps are chosen small enough to ensure convergence. We explain our choice of spatial and temporal discretisation in more detail in the text.

p. 10 line 7: What is a 'classical fracture'?
A classical fracture in the literature is a fracture enlarged by dissolution and experiencing a breakthrough event. The reason is the positive feedback between flow and dissolution. Will be explained in the text.

p.10 line 27: Why increase the tube diameter for the deeper fracture? As noted by Kaufmann et al. (2014) and many others, fracture permeability is expected to decrease with depth. It would seem more physically relevant to include a tube with the same diameter at the surface and decreasing with depth than a larger fixed-diameter tube.
The referee is correct, fracture width in general diminishes with depth. However, our reasoning to use a larger initial diameter for this case is mentioned in the text, we want to obtain a similar breakthrough time for a better comparison to the water-table model discussed earlier.

p. 12 line 25: Can you give an example when one might expect to have such a supersaturated influent solution?
We will add an example.

---

## Author Comment (AC2) · 30 Sep 2016

Reply to interactive comment by *Anonymous Referee #2*

We would like to thank the Anonymous Referee #2 for his valuable comments and take the opportunity to discuss the points made.

**Major comments**

1. The introduction and literature review appear to be disconnected from the main focuses of this study. For example, the paragraph that starts with line 3 on page 2 elaborates on several modeling work, but it is unclear how this study is related to or different from those studies, except for the Kaufmann et al (2014) study. Moreover, while the literatures summarized in section 2.2 showed how fracture evolution is affected by different flow regimes and other factors, most of these factors are not addresses or

are simplified in this study. For example, a drop of hydraulic head of 10 m was used in the manuscript, but it was not compared with realistic cases, and it is unclear why this specific flow regime is chosen. Furthermore, these literatures focus on small scale processes, while this study investigates field scale phenomena. Can the authors comment on how these literatures are relevant (or irrelevant) to this study and the scaling issue?.

We will move the review of selected work on fracture evolution to the introduction, group them into laboratory studies and numerical studies, and summarise their main outcomes as a motivation for our work.

2. Some aspects of the model framework and parameterization need to be clarified.
a. Page 4 line 10: 'with a fracture roughness coefficient mimicking small-scale wall irregularities in the fracture', is this roughness coefficient used in the calculation of reaction rate or flow or both?
For flow, we will add this in the text.

b. Page 5 line 13: what is the threshold Re used in this study?
$Re_c = 2200$ in our study, we will mention this explicitly.

c. Page 5: the friction factor for laminal flow was presented in eqn(3) but not used in eqn (1), was it used at all?
We condense the flow to a single non-linear equation, which has also been suggested by anonymous referee #1, then $f_l$ is used explicitly.

d. Page 5: how is the wall roughness (w in eqn(3)) defined and determined in this study and what is the impact of this parameter?
We add the definition to table 3.

e. Page 6: eqn (5) is very different from the advection-diffusion-reaction equation, even if the diffusion term is excluded. Can the authors comment on this and clarify the underlying assumptions? For example i. Is steady-state assumed, although it appears not to be the case given the following results?

Simple mass balance determined by flow rate and flux rate, we will add a reference pointing to the derivation.

ii. Is CFL criterion assumed to be one?.

We discretise the spatial coordinates as variable concentration increments (see also reply to referee #1), thus obeying a convergence criterium.

f. Page 8/29 (table 1): i. Only one kinetic coefficient is reported for the calcite reaction, but three reaction pathways were listed in (6), can the authors comment on this discrepancy?

The analytical solution for $c_{eq}$ derived by Dreybrodt uses the chemical reactions listed to approximate a closed-form solution for $c_{eq}$. All three surface reactions are therefore considered.

ii. For the gypsum reaction, the kinetic coefficients for the linear and non-linear rate laws are about one order of magnitude different according the reference cited, but the authors used the same kinetic coefficient, why and how the results may be affected?

We will add the relation for the non-linear rate constant $k_2$.

iii. The texts pointed out that different parameters are used in precipitation from dissolution, it should be clarified in table one.

We use the negative of the linear rate constant of each mineral species for precipitation, will be stated in the text.

g. Page 9 line 17: step 6 what is the time step?

Time stepping will be discussed in the text (see also referee #1).

h. Given the strong dependence of the evolution profile on kinetic rate laws, some sensitivity analysis or discussion of the uncertainties of the kinetic laws and coefficients should be provided.

There is a significant difference in evolution between the different soluble rock types considered, but the stated rate laws for each mineral species do only vary by around

one order of magnitude. Sensitivity analyses of the parameter values concerning the rate laws exist in the literature.

**Minor comments**

1. Page 2: how is section 2.1 fracture widening different from fracture dissolution discussed in section 2.2?
We will drop the sub-section titles, condense the processes section, and rearrange the literature review to the introduction. See also answer for referee #1.

2. There are a couple of typos. For example, Page 4 line 3 should be ?Jones and Detwiler (2016)?, and ?where? should be ?were?
Will be corrected.